# Quantitative Analysis of Seed Surface Tubercles in *Silene* Species

**DOI:** 10.3390/plants12193444

**Published:** 2023-09-29

**Authors:** José Luis Rodríguez-Lorenzo, José Javier Martín-Gómez, Ana Juan, Ángel Tocino, Emilio Cervantes

**Affiliations:** 1Plant Developmental Genetics, Institute of Biophysics v.v.i, Academy of Sciences of the Czech Republic, Královopolská 135, 612 65 Brno, Czech Republic; rodriguez@ibp.cz; 2Instituto de Recursos Naturales y Agrobiología, Consejo Superior de Investigaciones Científicas, Cordel de Merinas 40, 37008 Salamanca, Spain; jjavier.martin@irnasa.csic.es; 3Departamento de Ciencias Ambientales y Recursos Naturales, Universidad de Alicante, 03690 Alicante, Spain; ana.juan@ua.es; 4Departamento de Matemáticas, Facultad de Ciencias, Universidad de Salamanca, Plaza de la Merced 1-4, 37008 Salamanca, Spain; bacon@usal.es

**Keywords:** Bézier curve, Caryophyllaceae, complexity, curvature, development, seed surface, tubercle

## Abstract

In the Caryophyllaceae, seed surfaces contain cell protrusions, of varying sizes and shapes, called tubercles. Tubercles have long been described in many species, but quantitative analyses with measurements of size and shape are lacking in the literature. Based on optical photography, the seeds of *Silene* were classified into four types: smooth, rugose, echinate and papillose. Seeds in each of these groups have characteristic geometrical properties: smooth seeds lack tubercles and have the highest values of circularity and solidity in their lateral views, while papillose seeds have the largest tubercles and lowest values of circularity and solidity both in lateral and dorsal views. Here, tubercle width, height and slope, maximum and mean curvature values and maximum to mean curvature ratio were obtained for representative seeds of a total of 31 species, 12 belonging to *Silene* subg. *Behenantha* and 19 to *S*. subg. *Silene*. The seeds of the rugose type had lower values of curvature. Additionally, lower values of curvature were found in species of *S.* subg. *Silene* in comparison with *S.* subg. *Behenantha*. The seeds of *S*. subg. *Behenantha* had higher values of tubercle height and slope and higher values of maximum and average curvature and maximum to mean curvature ratio.

## 1. Introduction

Caryophyllaceae Juss. contains about 100 genera and ca. 2000 species characterized by anatropous to campylotropous ovules [1] and a peripheral position of the embryo in the developing seed [2]. Although single-seeded fruits seem to be the primitive characteristic in the family [3,4], most of the genera in the Alsinoideae and Caryophylloideae subfamilies have dry-dehiscent multi-seed capsules as their fruits [3]. The seed surfaces present micromorphological features such as cell perimeter and tubercle shape. These attributes have been used for species description and taxonomic purposes in diverse genera [5,6,7,8,9,10,11,12,13,14,15]. 

*Silene* L., with ca. 800 species, is the largest genus in the Caryophyllaceae. *Silene* seeds have a characteristic shape in the lateral view that may be described and quantified by comparison with a cardioid and derived figures [16,17,18]. 

Surface characteristics of *Silene* seeds have been traditionally analyzed via Scanning Electron Microscopy (SEM). Protuberances of cells at the seed surface are called tubercles, and early work indicated that they could be absent or present, and, according to their properties, the protuberances provided the basis to classify seeds into five types [19]. Another work highlighted two important aspects in the analysis of surface morphology in *Silene* seeds. The first concerns the level of conservation of the type between populations of the same species. Infraspecific variation between different populations was found in *S. italica* (L.) Pers., *S. spinescens* Sm., *S. nutans* L. and *S. coutinhoi* Rothm. & P.Silva [19]. This aspect was also found in *Geocarpon uniflorum* (Walter) E.E. Schill. (referred to as *Arenaria uniflora* (Walter) Muhl. in [7]), and in *Silene*, it was addressed more recently by Tabaripur et al. [20], who reported infraspecific variation in *S. odontopetala* Fenzl. 

A second aspect of importance involves the nomenclature of the tubercle shape diversity. The descriptors used by Ghazanfar for their description (round, cylindrical, conical, clavate, umbonate) had different success in later work by other groups based on SEM images. The group of Yildiz [21,22] retained the use of the terms, rounded, cylindrical and conical, replacing umbonated with mamillated and indicating that this characteristic is common in *S*. sect. *Siphonomorpha* as well. Ocaña et al. [23] distinguished between tubercles and papillae, the latter corresponding to umbonate tubercles of Ghazanfar and to mammillae of Dadandi and Yildiz. The group of Abbas Gholipour classified the seed coat as papillose, tuberculate or smooth [24], similar to the categories of Hoseini et al. [25]: flat, mammalian, echinate and tuberculate, where smooth means flat and papillose is equivalent to mammalian. Both the study of variation and the precise description of tubercles will benefit from the application of quantitative methods of tubercle analysis.

SEM photographs were the source with which to classify seeds by the shape of the tubercles at their surface in several species [26,27,28,29,30,31,32,33,34]. Nevertheless, optical photographs can be useful for seed surface description, and their results can be comparable with those of SEM. The observation of optical photographs of *Silene* seeds allowed us to differentiate between four groups, based on the tubercle structure: smooth, rugose, echinate and papillose [35,36]. Associated with these groups were differences in morphological measurements; thus, smooth seeds, without tubercles, had the highest values of circularity and solidity in their lateral views, while papillose seeds, with the largest tubercles, had the lowest values of circularity and solidity both in the lateral and dorsal views [35,36]. 

In the scientific literature reviewed here, the description of tubercles is based on qualitative features, but a quantitative description based on their measurements is lacking. A direct quantitative analysis of tubercle geometrics based on their width, height and curvature values is given for the first time in this work. Curvature represents the rate at which the unit tangent vector is changing with respect to arc length [37,38,39,40,41]. Curvature measurements are based on previous methods developed for roots analysis in *Arabidopsis* Heynh. in Holl & Heynh. (Brassicaceae) [37,38], as well as in wheat and grape seeds [39,40]. Curvature was measured in *Silene* seeds [41] and curvature values in the tubercles of *S. chlorifolia* Sm., a representative of rugose seeds, were compared with those in a population of *S. latifolia* Poir. Maximum and mean curvature values were higher in *S. latifolia*, suggesting that higher curvature values may be a property of echinate seeds in contrast with rugose seeds. In this work, we have analyzed the width and height of the tubercles, as well as curvature values in 31 representative species of *Silene*. The measurements of width, height and maximum and mean curvature revealed differences between rugose and echinate groups, as well as between the subgenera, *Silene* and *Behenantha*, and provide a basis for the quantitative analysis of tubercles and the relationship of tubercle type and taxonomic position.

## 2. Results

The results are organized into four subsections. The first one contains the comparison between rugose- and echinate-type seeds for all the tubercle measurements. These include width, height, slope, maximum curvature, mean curvature and the ratio of maximum and mean curvature for representative tubercles in 21 and 10 species of the rugose and echinate groups, respectively. The second and third subsections contain the comparison between species of each seed type, describing examples of the measurements in representative species. Finally, the fourth subsection contains the comparison of tubercle measurements between *Silene* species belonging to subg. *Behenantha* and *Silene*.

### 2.1. Comparison between Rugose and Echinate Types

On the basis of 21 species characterized by rugose seeds and 11 species with echinate seeds, the width at the base, height and slope, maximum and average curvature values and the ratio between maximum and average curvature values were calculated for a total of 393 tubercles. For each species, the number of seeds was between 1 and 3, and a total of 49 seeds were analyzed (see Figure A1 in Appendix B). ANOVA was carried out to compare all these measurements between the rugose and echinate types (Table 1).

Significant differences (*p* < 0.05) were found for all measurements with higher values for tubercle width at the base in the rugose seeds, and higher values for all the other measurements in the echinate seeds.

### 2.2. Comparison between Rugose Species

Table 2 contains the results of the ANOVA for the comparison of tubercle measurements between the 21 species belonging to the rugose seed type. Significant differences (*p* < 0.05) were found for all measurements. Tubercle width was between 45.0 (*S. inaperta*) and 138.7 μ (*S. marizii*), and tubercle height was between 9.5 (*S. inaperta*) and 28.0 μ (S. acutifolia). Tubercle slopes varied between 28.0 (*S. marizii*) and 63.4 (*S. spinescens*). Maximum curvature values were between 23.7 (*S. integripetala*) and 85.2 μ^−1^ (*S. spinescens*), with most of the values in the range of 20–50 μ^−1^. Mean curvature values were in the range of 12.7 (*S. nocturna*) to 48.1 μ^−1^ (*S. spinescens*). Maximum/mean curvature ratios were between 1.2 (*S. caryophylloides*) and 2.5 (*S. squamigera* subsp. *vesiculifera*), and only two species had values above 2: *S. nocturna* and *S. squamigera* subsp. *vesiculifera*. Few species had differences in curvature either for maximum or mean curvature values. Differences related to maximum curvature were detected between the group of lower values formed by *S. bupleuroides*, *S. chlorantha*, *S. chlorifolia*, *S. foliosa*, *S. frivaldskiana*, *S. integripetala*, *S. koreana*, *S. marizii* and *S. nocturna* and the group of species with higher values (*S. otites*, *S. spinescens* and *S. squamigera* subsp. *vesiculifera*). In relation to mean curvature, lower values were found in *S. nocturna* and higher values in *S. otites* and *S. spinescens*. Finally, the ratio between maximum and average curvatures showed lower values in *S. caryophylloides*, *S. dinarica* and *S. integripetala*, and higher values in *S. squamigera* subsp. *vesiculifera*.

The following sections describe the results obtained for six tubercles from representative species in this group. The data are grouped in three parts, corresponding to species with the lowest values of maximum curvature (*S. bupleuroides*, *S. chlorantha*, *S. frivaldskiana*, *S. integripetala*), species with the highest values of maximum curvature (*S. spinescens* and *S. squamigera* subsp. *vesiculifera*), and a separate section for *S. gigantea*, a species that presented two types of tubercles. The main aspects observed are indicated, as well as when these are shared by other species. The differences between the data presented in Table 2 and those in the subsections below are because the following subsections contain data for the first six tubercles measured for each species. The Mathematica^®^.nb files are provided as Appendix A (See Appendix A).

#### 2.2.1. Rugose Species with Lower Maximum Curvature Values

*S. bupleuroides* Kom.

The tubercles examined of *S. bupleuroides* are rather low and flat, with low values of slope, and their maximum curvature values are deviated from a central position (Figure 1). Table 3 contains the values for tubercle width, height, slope, maximum and average curvature for six tubercles. Tubercle 1 showed the lowest values of height and slope (15 and 34.5, respectively), whereas the highest values were found for tubercle 4 (23 and 51.7, respectively). The maximum values for curvature and mean curvature corresponded to tubercle 2 (45.7 and 28.5, respectively).

2.*S. chlorantha* (Willd.) Ehrh.

The tubercles of *S. chlorantha* are irregular in size and shape, resembling deformed arcs of circumference (Figure 2). Their curvature plots resemble horizontally placed curly brackets with multiple maximum values located in both central and lateral positions. Table 4 contains the values for tubercle width, height, slope, maximum and average curvature for each of these tubercles. Tubercles 4 and 2 showed the lowest and highest values for height and slope, respectively. Changes in the direction of the slope were observed in most of the tubercles.

3.*S. frivaldskiana* Hampe.

The tubercles studied of *S. frivaldskiana* are rather low and smooth, with low values of slope, and maximum curvature values maintained through most of their length in a central position (Figure 3 and Table 5). The measured features are, in general, quite similar among the tubercles, with the maximum differences in width found between tubercles 4 and 3 (63 and 83, respectively), in height and slope for tubercles 2 and 4. The ratio maximum/mean curvature is low (ranging from 1.2 to 1.6), indicating that both values are quite similar. Tubercles 4 and 5, with maximum and average curvature values close to 20, resemble the arc of a circumference of radium inverse to the curvature value, in this case, of 50 μ approximately.

4.*S. integripetala* Bory & Chaub.

Figure 4 contains the lateral view of a seed of *S. integripetala* with the indication of the tubercles analyzed in this section (numbered from 1 to 6), and the variation of curvature. The data in Table 6 reveal a gradation in their shape, from a central apical point with maximum curvature (in tubercle 1), to more plane-like (tubercles 2, 4, 5 and 6) and finally, a tubercle with two major curvature points at the sides (tubercle 3). Additionally, tubercle 1 is characterized by the lowest width and height, and tubercle 3 is the highest, with the maximum slope values.

As with most tubercles in this section, the ratio maximum to average curvature is low (1.4), indicating that both values are similar along the tubercle length and, thus, that the tubercle resembles the arc of a circumference of radium inverse to the mean curvature value, in this case, between 40 and 50 μ.

#### 2.2.2. Rugose Species with High Curvature Values

*S. spinescens* Sm.

Figure 5 presents a seed of *S. spinescens* in the lateral view. Tubercles are numbered from 1 to 6. Table 7 contains the values for tubercle length, height, slope, maximum and average curvature for each of these tubercles. This seed provides an example of two types of tubercles: rounded, typically of the rugose type (tubercle numbers 1, 4, 5 and 6) and more pointed, with higher curvature values, tending towards echinate (2 and 3). The measured values also support this differentiation since the maximum curvature values were observed for tubercles 2 and 3.

2.*S. squamigera* Boiss. subsp. *vesiculifera* (J.Gay ex Boiss.) Coode and Cullen

Figure 6 presents a seed of *S. squamigera* subsp. *vesiculifera* in the lateral view. Tubercles are numbered from 1 to 6. No clear acute tubercles were observed, and the curvature shape is mostly rounded. However, tubercles 3, 4, 5 and 6 were characterized by the presence of a certain type of secondary excrescence, the presence of which appears to affect to the maximum curvature variable (Table 8). The remaining tubercles have lower curvature values.

#### 2.2.3. Rugose Species with Two Types of Tubercles

*S. gigantea* (L.) L.

Figure 7 contains a seed of *S. gigantea* in the lateral view. Tubercles are numbered from 1 to 6. Table 9 shows the values for tubercle length, height, slope and maximum and average curvature for each of these tubercles. The analysis of the curvature, based on Bézier curves, revealed two morphological types of tubercles: (i) rounded, with lower curvature values (tubercles 1 to 4) and (ii) acute, with a peak at the point of maximum curvature (tubercles 5 and 6). These latter two tubercles resemble those typical of the echinate type. The highest values for most the measured variables (H, S, maximum curvature and ratio maximum curvature/mean curvature) were found for tubercles 5 and 6, but for width, tubercle 2 (Table 9).

### 2.3. Comparison between Echinate Species

Table 10 contains the results of the ANOVA for the comparison of tubercle measurements among 10 species belonging to the echinate seed type. The differences between species were significant for all measurements. Tubercle width was between 42.2 (*S. yunnanensis*) and 91.2 microns (*S. samojedorum*), and tubercle height was in the range of 23.7 (*S. longicilia*) to 83.3 (*S. petersonii*). Tubercle slope varied between 67.3 (*S. longicilia*) and 278.5 (*S. yunnanensis*). Maximum curvature values were between 62.6 (*S. dichotoma*) and 219.2 (*S. samojedorum*). Mean curvature values were in the range of 20.5 (*S. caroliniana*) to 91.5 (*S. samojedorum*). The maximum/mean curvature ratio was between 1.6 (*S. dichotoma*) and 6.1 (*S. caroliniana*). There were large differences between species for all measurements.

According to the maximum curvature values, two groups were distinguishable, corresponding, respectively, to species with lower and higher maximum curvature values. The group of higher maximum curvature values comprised *S. behen*, *S. samojedorum* and *S. yunnanensis*, while the group of lower maximum curvature values comprised *S. ciliata*, *S. dichotoma* and *S. longicilia*.

According to the mean curvature values, two groups were distinguishable, corresponding, respectively, to species with lower and higher mean curvature values. The group of higher mean curvature values comprised *S. fabaria*, *S. petersonii*, *S. samojedorum* and *S. yunnanensis*, while the group of lower mean curvature values comprised *S. aprica* subsp. *oldhamiana*, *S. caroliniana*, *S. ciliata*, *S. dichotoma* and *S. longicilia*.

Finally, the ratio between maximum and average curvatures revealed two groups comprising *S. caroliniana* (higher value) and the group formed by *S. ciliata*, *S. dichotoma*, *S. fabaria*, *S. longicilia* and *S. samojedorum*.

The following sections describe the results of representative species in the above groups: *S. aprica* subsp. *oldhamiana*, *S. caroliniana S. ciliata* (lower maximum curvature) and *S. yunnanensis* (higher maximum curvature values).

#### 2.3.1. Echinate Species with Low Maximum Curvature Values

*S. aprica* subsp. *oldhamiana* (Miq.) C.Y.Wu ex C.L.Tang.

Figure 8 presents a seed of *S. aprica* subsp. *oldhamiana* in the lateral view. The tubercles analyzed are numbered from 1 to 6. The six studied tubercles are notably triangular, with a well-defined central peak and a maximum curvature between 81 (tubercle 3) and 168 (tubercle 2) (Table 11). Although the minimum values for the studied variables were observed for different tubercles, the maximum data mostly corresponded to tubercles 2 and 5. The observed values of the ratio of maximum/mean curvature, up to 5.2, illustrated the notable difference between the maximum curvature and the mean curvature of the tubercles.

2.*S. caroliniana* Walter

Figure 9 shows a seed of *S. caroliniana* in the lateral view. The tubercles analyzed in this section are numbered from 1 to 6. For this species, the studied tubercles showed a well-visible central peak but they were remarkably different, with a gradation of their shape, from a flattened (2), to clearly acute (4,5,6) tubercles. These differences were supported by the measured variables (Table 12), since the minimum values mostly corresponded to tubercle 2, but the highest value to tubercle 4. The one exception was related to the width of the tubercles, with tubercle 1 having the maximum values (79) and tubercle 3 the minimum ones (59).

3.*S. ciliata* Pourr.

Figure 10 shows a seed of *S. ciliata* in the lateral view. The tubercles analyzed in this section are numbered from 1 to 6. The studied tubercles had a similar shape with a central peak, with the maximum curvature values between 41 (tubercle 5) and 83 (tubercle 6) (Table 13). For most of the studied variables, tubercles 3 and 5 showed the lowest values, whereas the highest ones corresponded to tubercles 2 and 6.

#### 2.3.2. Echinate Species with Higher Maximum Curvature Values

*S. yunnanensis* Franch

Figure 11 contains a seed of *S. yunnanensis* in the lateral view. Tubercles are numbered from 1 to 6. The studied tubercles were quite similar—clearly acute with a central peak, with no noticeable differences in their shape. Tubercle 1 showed the lowest values of height and slope, and the maximum values corresponded to tubercles 2 and 3 (Table 14). The maximum curvature data were found for tubercles 5 and 6 (341.2 and 234.2, respectively), with values notably different from the remaining tubercles.

### 2.4. Relationship between Tubercle Type and Subgenera

The results show a relationship between the measurements of the tubercles and the subgenera. Table 15 contains the comparison of the mean values of width, height, slope, maximum curvature, mean curvature and the ratio of maximum to mean curvature between seeds of *Silene* species belonging to *S.* subg. *Behenantha* and *S.* subg. *Silene*. The values were higher in subg. *Behenantha* for all the measurements, except tubercle width.

The dendrogram in Figure 12 shows the relationship between *Silene* species based on the combination of values for all six measurements indicated in Table 15. Most species of *S.* subg. *Behenantha* grouped in the lower part of the diagram had higher curvature values.

## 3. Discussion

### 3.1. Seed Surface Morphology in Silene Species

The analysis of *Silene* seed surface morphology has been traditionally based on Scanning Electron Microscopy (SEM) because this technique gives a clear definition of cell boundaries, tubercle shape, their distribution and, hence, aspects of the seed surface. Greuter [34] stated that cell shape, cell surface granularity and the presence/lack of mamilla could be used as diagnostic features for his proposed infrageneric classification of *Silene*. As an example, and in relation to the morphology of the tubercles, Ghazanfar [19] described them as rounded, cylindrical, clavate, umbonate or conical; Dadandi and Yildiz [21] as rounded, cylindrical, conical and mamillated. Runyeon and Prentice [33] defined the seed surface as tubercled or smooth for diverse populations of *Silene vulgaris* and *S. uniflora*, and more recently, Hosseini et al. [25] expanded seed surface morphological characterization using terms such as mammalian, echinate, tuberculate and flat. Although there are some differences in the uses of specific nomenclatures, the results may be comparable between them and with those obtained via optical imaging.

Recent studies based on optical imaging classified *Silene* seeds into four morphological types named smooth, rugose, echinate and papillose [35,36]. Out of 100 species analyzed, the majority were classified as rugose and echinate, with 16 species being classified as smooth and only 4 species as papillose. Most of the smooth seeds belonged to *S*. subg. *Silene* sect. *Silene*, and the four species with papillose seeds belonged to *S*. subg. *Behenantha* sect. *Physolychnis* (*S. laciniata* Cav. and *S. magellanica* (Desr.) Bocquet), to *S*. subg. *Behenantha* sect. *Behenantha* (*S. holzmannii* Heldr. ex Boiss.) and to *S.* subg. *Silene* sect. *Silene* (*S. perlmanii* W.L.Wagner, D.R.Herbst & Sohmer). The geometric characteristics of smooth and papillose seeds allow them to be distinguished from among each other and from other seed groups due to (i) the lack of tubercles in the case of smooth seeds and (ii) for papillose seeds, the tubercles or papillae showing a varied shape and uneven size, characterized by lower values of circularity and solidity [35,36].

According to previous studies based on optical seed images [35,36], rugose tubercles are typically rounded and echinates are more acute or triangular. This difference suggests that these two types of tubercles could be also characterized by quantitative geometric properties. A series of measurements presented in this work allow them to be differentiated from each other. Both types of seed tubercle may be thoroughly differentiated via curvature analysis. Curvature is the magnitude that indicates the rate of change in the slope through the points of a curve [37,38,39,40,41]. This has a constant value in a circumference, which equals the inverse of its radius with units of length^−1^. The maximum to mean curvature ratio gives a measure of the separation from the figure of an arc of circumference. Both maximum and mean curvature values and maximum to mean curvature ratios were significant higher for echinate tubercles than for rugose ones. Tubercle curvature, as with any other tubercle measurement, is a geometric feature of tubercles, not of seeds. These curvature values could adequately characterize the geometric differences between them. In addition, the combination of a smaller width, with a higher height and notably higher values of slope might also support the definition of echinate tubercles in contrast to rugose.

*Silene* cell surface projections may vary infraspecifically (e.g., *S. viscosa* Pers., see [25]). In addition, it may be possible to find diverse types of tubercles within the same seed. Our data demonstrated that the presence of two types of tubercles was relatively frequent, as it was observed for *S. spinescens* and *S. gigantea*. Rugose tubercles were well characterized by the highest values of width and the lowest values for the remaining measurements in comparison to echinate tubercles. Therefore, the geometric variables would be very useful in distinguishing and characterizing the morphology of tubercles from quantitative data, as opposed to only via qualitative description. The presence of additional excrescences on the top of rugose tubercles, as it was observed for *S. squamigera* subsp. *vesiculifera*, can change the typical values of the curvature variables of a rugose tubercle.

The physiological function of the tubercles and their adaptive meaning is still a matter of debate, and an association between tubercle types and particular environmental factors has not been conclusively proven. The tubercles may be involved in energy homeostasis and as mechanical factors facilitating seed transport [42]. Increased tubercle size means a higher surface area to volume ratio with the consequent increase in energy and gas exchange [43]. Additionally, tubercles may have a role for anchoring seeds in the rocky soil of cliffs [44,45,46].

There is no clear adaptive benefit of tubercle shape in relation to geographic localization or climatic conditions. Based on the observation of 167 collections of wild seeds and 97 sets of mature greenhouse-grown progeny, Prentice analyzed the variation in tubercle shape for *S. latifolia* and *S. dioica*, concluding that seeds with conical tubercles have a more eastern localization than those with rounded tubercles [47]. In another study with herbarium material, the analysis of 216 seeds from diverse origins [48] allowed the researchers to conclude that while seeds from cold-winter regions have tall tubercles, seeds from warm-winter regions possess, in general, shorter tubercles.

Concerning the relationship between tubercle type and taxonomic subgenera, our results show an association of the rugose tubercle type with *subg*. *Silene* and the echinate type with *Behenantha*. Although in many instances, morphological characteristics appear independently in separate lineages over the course of evolution (homoplasy) [19,21,25,27,36], the quantitative analyses presented here support the predominance of the rugose tubercle type, characterized by lower curvature values, in the species of *S.* subg. *Silene* and the echinate type, of higher curvature values, in S. subg. *Behenantha*. The analysis may be applied to other species and populations to confirm whether this possibility has a general application.

### 3.2. Seed Surface Morphology in the Caryophyllaceae

Several articles have been dedicated to seed micromorphology in species other than *Silene* in the Caryophyllaceae [5,6,7,8,9,10,11,12,13,14,49,50,51,52,53,54,55,56,57,58,59,60,61,62,63,64]. In some instances, tubercle shape is associated with a given phylogenetic group or a particular geographic location [6,10,49,50,51]. The seeds of *Moheringia* L. species with long tubercles (papillate) have been described in warm-winter regions in the Eastern Balkans and the Iberian Peninsula [10]. In *Schiedea* Cham. & Schltdl., papillate margins are found in species occurring in ecologically extreme environments, such as wet forests of the higher elevations (*S. trinervis*) or dry slopes (*S. adamantis*) [50].

Finally, micromorphological characters have been recently considered as possible taxonomic criteria in other families of the order, Caryophyllales, such as the Aizoaceae and the Molluginaceae [65,66]. The seeds of *Portulacca*, in the Portulacaceae, have tubercles resembling those of *Silene* [67,68], which can be accurately measured using the methods described here.

There is a long-standing controversy about the applicability of tubercles in taxonomy that may benefit from the quantitative methods described here.

## 4. Materials and Methods

### 4.1. Silene Seeds

The populations of seeds analyzed in this work are listed in Table 16. Representative seed images from populations labelled JBUV can be observed and downloaded at https://zenodo.org/record/7330942#.Y3 Y8 Hn3 MJD8 (accessed on 26 September 2023). Seed images of the populations of *S. acutifolia* and *S. otites* are available at https://zenodo.org/record/4035649 (accessed on 26 September 2023). Seed images for the remaining populations (*S. behen*, *S. caroliniana*, *S. inaperta* and *S. nocturna*) are available under request.

**Table 16 plants-12-03444-t016:** List of seed populations analyzed in this work. The populations labelled JBUV were obtained from the carpoespermateca at the Botanical Garden of the University of Valencia and originate from an exchange protocol between seed collections through the world. The name of the subgenus and the corresponding section (between brackets) are indicated. “u.” means unknown place of origin. The ascription of each species to subgenera and sections is taken from [69].

Species	Type (Rugose or Echinate)	Received from	Registered by (Place of Origin)	Subgenus (Section)
*Silene acutifolia* (L.) Link ex Rohrb.	R	IBP collection Acad Sci CzR, Brno	IBP collection (u.)	*Behenantha (Acutifoliae)*
*Silene aprica* subsp. *oldhamiana* (Miq.) C.Y.Wu ex C.L.Tang	E	JBUV333	Chollipo Arboretum BG, South Korea (u.)	*Behenantha* *(* *Physolychnis* *)*
*S. behen* L.	E	IBP collection Acad Sci CzR, Brno	IBP collection (u.)	*Behenantha* *(* *Behenantha* *)*
*S. bupleuroides* L.	R	JBUVC2787	u. Hortus Botanicus Vacratot, Hungary (u.)	*Silene* *(* *Sclerocalycinae* *)*
*S. caroliniana* Walter	E	IBP collection Acad Sci CzR, Brno	IBP collection (u.)	*Behenantha* (*Physolychnis*)
*S. caryophylloides* (Poir.) Otth	R	JBUV950	Botanischer Garten der Universität Tübingen (Holubec, Turkey, Ulu Dag)	*Silene* (*Auriculatae*)
*S. chlorantha* (Willd.) Ehrh.	R	JBUV1028	Botanischer Garten der Universität Potsdam Germany. (Brandenburg, Odergebiet, an der Bahn, SW Bahnhof Podelzig, Germany)	*Silene* (*Siphonomorpha*)
*S. chlorifolia* Sm.	R	JBUV1000	BG der Martin-Luther-Univ. Halle-Wittenberg, Germany (u.)	*Silene* (*Sclerocalycinae*)
*S. ciliata* Pourr.	E	JBUV219	Botany Hung. Acad. of Sciences (Botanical Garden)	*Silene* (*Silene*)
*S. dichotoma* Ehrh.	E	JBUV668	Botanic Garden of the University of Copenhagen, Denmark (u.)	*Behenantha* (*Dichotomae*)
*S. dinarica* Spreng.	R	JBUV1600	Botanicka Zahrada Teplice, Czech Republic (u.)	*Silene* (*Siphonomorpha*)
*S. fabaria* (L.) Coyte	E	JBUV3188	Botanischer Garten der Universität Bonn, Germany (Chakidiki, south of Ouranopolis, towards the border of Athos, Greece)	*Behenantha* (*Behenantha*)
*S. foliosa* Maxim.	R	JBUV2177	Vladivostok Botanical Garden, Russia (Gamow Peninsula)	*Silene* (*Siphonomorpha*)
*S. frivaldszkyana* Hampe	R	JBUV2881	Siberian Botanical Garden of Tomsk State University, Russia (u.)	*Silene* (*Siphonomorpha*)
*S. gigantea* (L.) L.	R	JBUV1494	Julia & Alexander N. Diomides Botanic Garden, Athens, Greece (Cult./ATHD)	*Silene* (*Siphonomorpha*)
*S. inaperta* L.	R	AJ270	Collection of Ana Juan (Rambla Bateig, Elda, Alicante, Spain)	*Silene* (*Muscipula*)
*S. inaperta* L.	R	AJ335	Collection of Ana Juan (Rambla de Caprala, Petrer, Alicante, Spain)	*Silene* (*Muscipula*)
*S. integripetala* Bory and Chaub.	R	JBUV1001	BG der Martin-Luther-Univ. Halle-Wittenberg, Germany	*Behenantha* (*Sedoides*)
*S. koreana* Kom.	R	JBUV1917	Botanic Garden of Perm State University, Russia	*Silene* (*Siphonomorpha*)
*S*. *linicola* C.C. Gmel.	R	JBUV1056	Jardin Botanique de la ville de Lyon Aube (10), entre Auxerre et Troyes, France.	*Silene* (*Lasiocalycinae*)
*S*. *longicilia* (Brot.) Otth	E	JBUV712	BG Universidade de Coimbra (Serra da Boa Viagem—Figueira da Foz, Portugal)	*Silene* (*Siphonomorpha*)
*S. marizii* Samp.	R	JBUV713	BG Universidade de Coimbra (Nespereira—Celorico da Beira, Portugal)	*Behenantha* (*Melandrium*)
*S. nocturna* L.	R	AJ316	Collection of Ana Juan (Sierra de Irta, Peñíscola, Castellón, Spain)	*Silene* (*Silene*)
*S. otites* (L.) Wibel	R	(IBP collection, Brno)	IBP collection (u.)	*Silene* (*Siphonomorpha*)
*S. paradoxa* L.	R	JBUV1142	Station Alpine du Lautaret. Univ. Joseph Fourier Défilé d’Inzecca (Corse, France, 250 m)	*Silene* (*Siphonomorpha*)
*S. petersonii* Maguire	E	JBUV1002	BG der Martin-Luther-univ. Halle-Wittenberg, Germany (Mount Brocken Garden)	*Behenantha* (*Physolychnis*)
*S. pygmaea* Adams	R	JBUV964	St Andrews Botanical Garden, Scotland (u.)	*Silene* (*Auriculatae*)
*S. regia* Sims	R	JBUV2630	Botanischer Garten Universität Hamburg Nachzucht BG Hamburg, Germany (Winona, MN/US; Prairie Moon Nursery, USA)	*Behenantha* (*Physolychnis*)
*S. samojedorum* (Sambuk) Oxelman	E	JBUV1069	Hortus Botanicus Patavinus, Italy	*Behenantha* (*Physolychnis*)
*S. spinescens* Sm.	R	JBUV1496	Julia & Alexander N. Diomides Botanic Garden (Spont./Sounion National Park-Attiki, Athens, Greece)	*Silene* (*Siphonomorpha*)
*S. squamigera* Boiss. subsp. *vesiculifera* (J.Gay ex Boiss.) Coode and Cullen	R	JBUV671	Botanic Garden of the University of Copenhagen, Danmark	*Silene* (*Lasiocalycinae*)
*S. yunnanensis* Franch.	E	JBUV2050	Botanischer Garten der Universität Zürich, Germany	*Behenantha* (*Cucubaloides*)

### 4.2. Seed Images

For the analysis of individual tubercles, photographs were taken with a Nikon Stereomicroscope Model SMZ1500 (Nikon, Tokio, Japan), equipped with a camera Nikon DS-Fi1 of 5.24 megapixels (Nikon, Tokio, Japan). Photographs were stored as JPG images of 2560 × 1920 with 300 ppp.

### 4.3. Tubercle Measurements

Tubercle measurements and curvature analyses were applied to seeds of 31 species of *Silene*. Most of them were obtained from the carpoespermateca at the Botanical Garden of the University of Valencia by an exchange protocol between seed collections through the world. Measurements of width, height, slope and curvature (maximum and average curvature values) were taken for a total of 396 tubercles (268 corresponding to rugose seeds and 128 to echinate seeds). The number of seeds was between 1 and 3 for each species, as indicated in Table 2 and Table 10 and Figure A1 in Appendix B.

#### 4.3.1. Width, Height and Slope Measurements in the Tubercles

Seed images containing a ruler of 1 mm were opened in Image J. Two direct measurements were made for each of the tubercles indicated in Figure 4: tubercle width at the base (*W*) and tubercle height (*H*). The slope of each tubercle (*S*) was calculated as
$$S=\frac{W}{H}*200$$

#### 4.3.2. Curvature Measurements

Maximum absolute values and average curvatures were determined for individual tubercles of each species according to established procedures [37,38,39,40,41] (See Appendix A). In the measurements of curvature for individual tubercles, the points were taken automatically with the function, Analyze line graph, of Image J. Between 6 and 22 tubercles of representative seeds (see Figure A1 in Appendix B) were analyzed for each species. Individual images (JPEG) were kept of each tubercle and vertically oriented. Images were opened in Image J and converted to 8-bit, threshold values were adjusted, and the image was analyzed. The curve corresponding to the tubercle was selected from the outlines, and a new threshold was defined before the corresponding line graph was analyzed, to obtain the x,y coordinates. The coordinates were the basis of obtaining the Bézier curve and the corresponding curvature values according to published protocols [37,38,39,40,41]. Curvature was given in micron^−1^ × (1000); thus, a curvature of 20 corresponds to a circumference of 50 microns (1/50 × 1000) and a curvature of 100 to a to a circumference of 10 microns.

### 4.4. Statistical Analysis

ANOVA was used to show significant differences between populations for the measured variables. In the case of the comparison of morphological characteristics involving more than two populations, ANOVA was followed by a Tukey test to obtain specific information on which means were significantly different from one another. Statistical analyses (ANOVA) were conducted with IBM SPSS statistics v28 (SPSS 2021).

The Euclidean distance and Ward algorithm for clustering were used to calculate the dendrogram. The matrix used for the analysis contained the data for maximum curvature values in Table 2 and Table 10.

## 5. Conclusions

A quantitative analysis of tubercles has been made here of seeds of *Silene*, belonging to 31 species that had previously been classified as rugose (21 species) or echinate (10 species). Of these, 19 belonged to *S.* subg. *Silene* and 12 to *Behenantha*. The analysis was based on tubercle dimensions and curvature. Width at the base, height and slope, together with maximum and average curvature values were obtained for 396 tubercles, in a total of 49 seeds corresponding to 21 species. Rugose tubercles had higher values of tubercle width and lower values for the other measurements, in comparison to echinate-type tubercles. The seeds of the rugose type had lower values of curvature. Additionally, lower values of curvature were found in species of *S.* subg. *Silene* in comparison with *S.* subg. *Behenantha*. The protocols applied in this article may be helpful for detailed morphological analyses, determining not only differences of tubercle type (rounded or rugose versus conical or echinate) but also more subtle quantitative changes in tubercle morphology.

## Figures and Tables

**Figure 1 plants-12-03444-f001:**
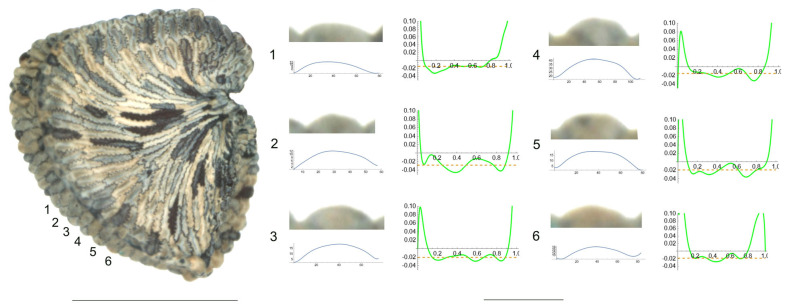
Left: lateral view of a seed of *S. bupleuroides* with the tubercles analyzed numbered 1 to 6. The bar on the left represents 1 mm. Right: individual view of the tubercles, Bézier curves (below) and curvature plots representing the variation in curvature along the Bézier curve. The discontinuous line in the curvature plot represents the value of mean curvature. Bar on the right side, below the tubercle images, represents 100 μ.

**Figure 2 plants-12-03444-f002:**
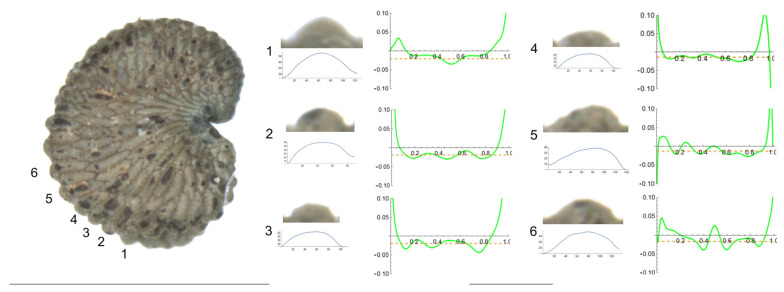
Left: lateral view of a seed of *S. chlorantha* with the tubercles analyzed numbered 1 to 6. The bar on the left represents 1 mm. Right: individual view of the tubercles, Bézier curves (below each tubercle) and curvature plots representing the variation in curvature along the Bézier curve. The discontinuous line in the curvature plot represents mean curvature. Bar on the right side, below the tubercle images, represents 100 μ.

**Figure 3 plants-12-03444-f003:**
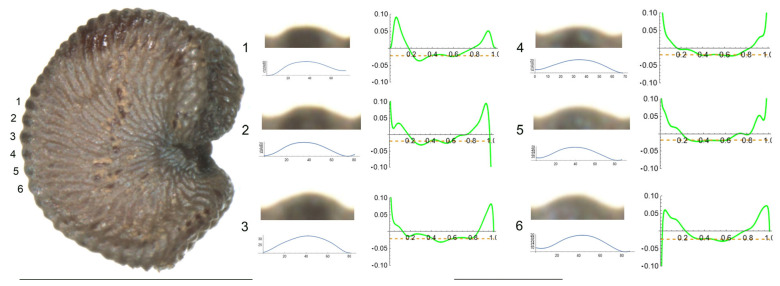
Left: lateral view of a seed of *S. frivaldskiana* with the tubercles analyzed numbered 1 to 6. The bar on the left represents 1 mm. Right: individual view of the tubercles, Bézier curves (below each tubercle) and curvature plots representing the variation in curvature along the Bézier curve. The discontinuous line in the curvature plot represents mean curvature. Bar on the right side, below the tubercle and curvature images, represents 100 μ.

**Figure 4 plants-12-03444-f004:**
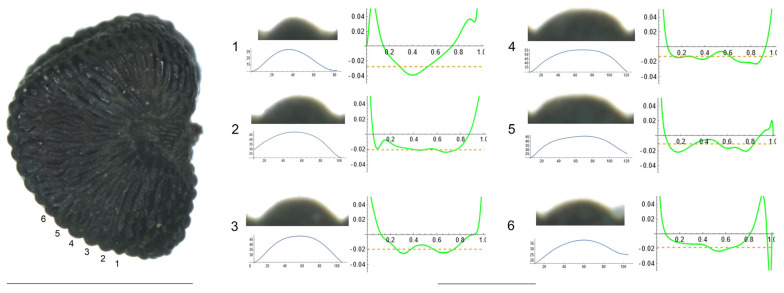
Left: lateral view of a seed of *S. integripetala* with the tubercles analyzed numbered 1 to 6. Bar represents 1 mm. Right: individual view of the analyzed tubercles, Bézier curves (below each tubercle) and curvature plots representing the variation in curvature along the Bézier curve. The discontinuous line in the curvature plot represents mean curvature. The bar below the tubercle and curvature images equals 100 μ.

**Figure 5 plants-12-03444-f005:**
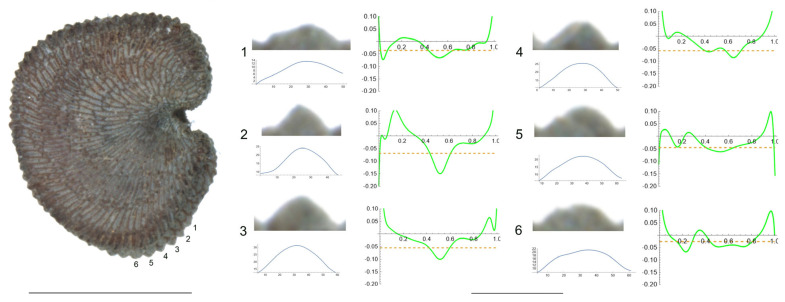
Left: Lateral view of a seed of *S. spinescens* with the tubercles object of analysis numbered 1 to 6. The bar on the left represents 1 mm. Right: individual view of the analyzed tubercles, Bézier curves (below each tubercle) and curvature plots representing the variation in curvature along the Bézier curve. The discontinuous line in the curvature plot represents the value of mean curvature. Bar on the right side below the tubercle images represents 100 μ.

**Figure 6 plants-12-03444-f006:**
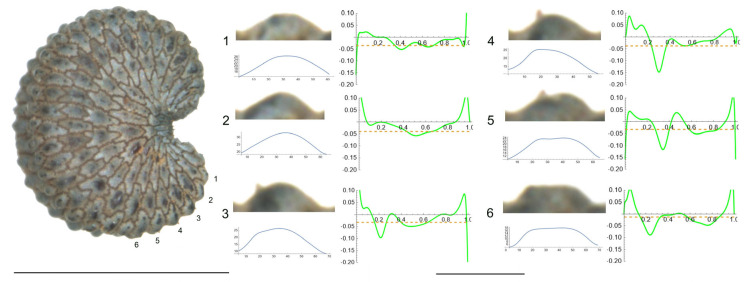
Left: lateral view of a seed of *S. squamigera* subsp. *vesiculifera* with the tubercles analyzed numbered 1 to 6. The bar on the left represents 1 mm. Right: individual view of the tubercles, Bézier curves (below each tubercle) and curvature plots representing the variation in curvature along the Bézier curve. The discontinuous line in the curvature plot represents the value of mean curvature. Bar on the right side below the tubercle images, represents 100 μ.

**Figure 7 plants-12-03444-f007:**
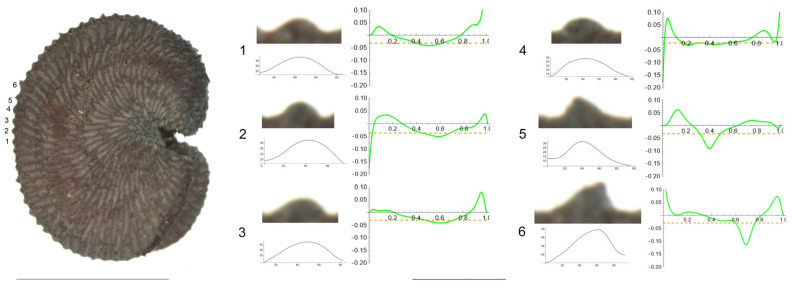
Left: lateral view of a seed of *S. gigantea* with the tubercles object of analysis numbered 1 to 6. The bar on the left represents 1 mm. Right: individual view of the tubercles, Bézier curves (below each tubercle) and curvature plots representing the variation in curvature along the Bézier curve. The discontinuous line in the curvature plot represents the value of mean curvature. Bar on the right side below the tubercle images represents 100 μ.

**Figure 8 plants-12-03444-f008:**
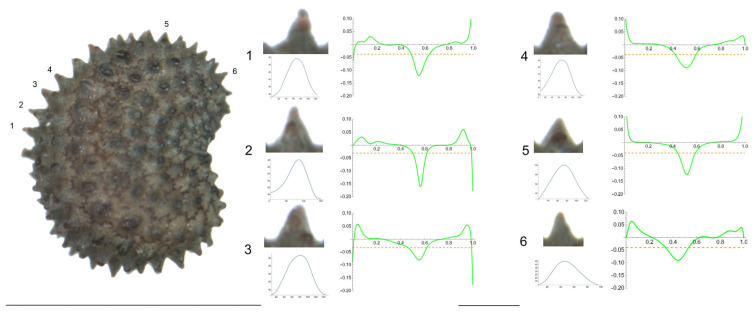
Left: lateral view of a seed of *S. aprica* with the tubercles object of analysis numbered 1 to 6. The bar on the left represents 1 mm. Right: individual view of the tubercles, Bézier curves (below each tubercle) and curvature plots representing the variation in curvature along the Bézier curve. The discontinuous line in the curvature plot represents the value of mean curvature. Bar on the right side below the tubercle images represents 100 μ.

**Figure 9 plants-12-03444-f009:**
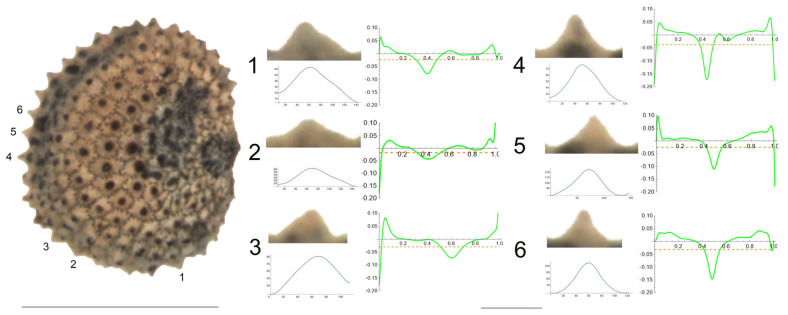
Left: lateral view of a seed of *S. caroliniana* with the tubercles analyzed numbered 1 to 6. The bar on the left represents 1 mm. Right: individual view of the tubercles, Bézier curves (below each tubercle) and curvature plots representing the variation in curvature along the Bézier curve. The discontinuous line in the curvature plot represents the value of mean curvature. Bar on the right side below the tubercle images represents 100 μ.

**Figure 10 plants-12-03444-f010:**
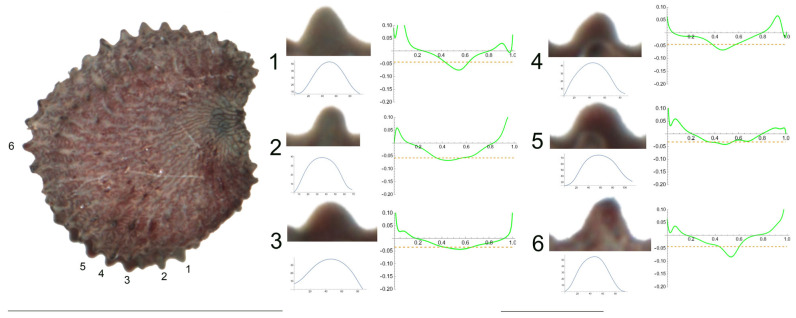
Left: lateral view of a seed of *S. ciliata* with the tubercles analyzed numbered 1 to 6. The bar on the left represents 1 mm. Right: individual view of the tubercles, Bézier curves (below each tubercle) and curvature plots representing the variation in curvature along the Bézier curve. The discontinuous line in the curvature plot represents the value of mean curvature. Bar on the right side below the tubercle images represents 100 μ.

**Figure 11 plants-12-03444-f011:**
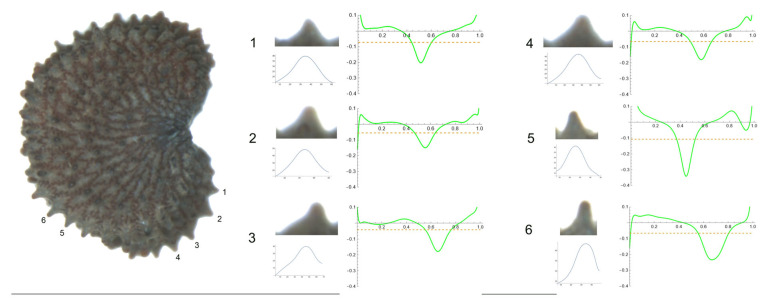
Left: lateral view of a seed of *S. yunnanensis* with the tubercles analyzed numbered 1 to 6. The bar on the left represents 1 mm. Right: individual view of the tubercles, Bézier curves (below each tubercle) and curvature plots representing the variation in curvature along the Bézier curve. The discontinuous line in the curvature plot represents the value of mean curvature. Bar on the right side below the tubercle images represents 100 μ.

**Figure 12 plants-12-03444-f012:**
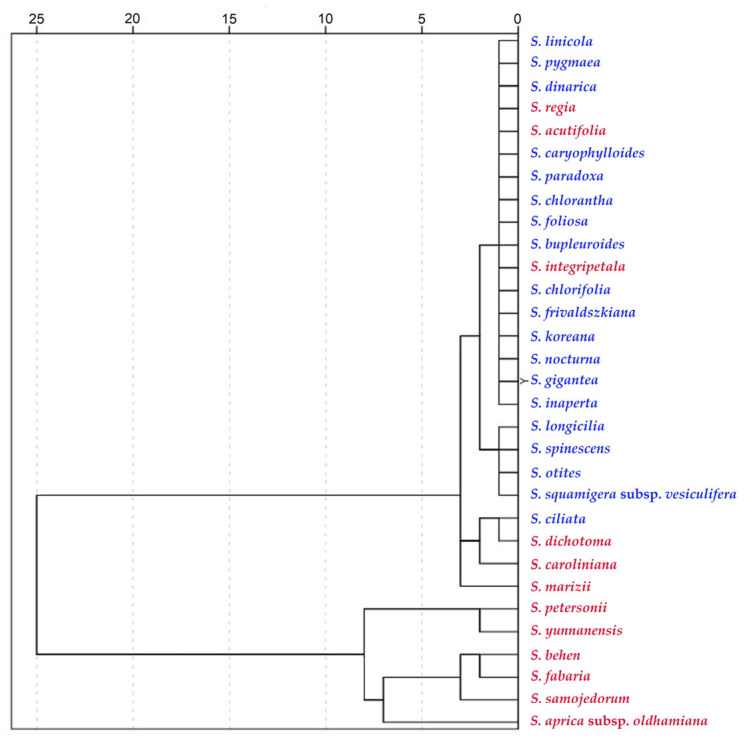
Dendrogram based on hierarchical clustering with the values of six measurements indicated in Table 15. Species of subg. *Behenantha* are concentrated in the lower part of the diagram, corresponding with the higher values of curvature, while species of subg. *Silene* are grouped in the upper branches. Results are shown for the 31 species within Table 16 (12 belonging to *S*. subg. *Behenantha* in red and 19 belonging to *S*. subg. *Silene* in blue).

**Table 1 plants-12-03444-t001:** Results of the comparison (ANOVA) between rugose and echinate seeds for tubercle width (W), height (H), slope (S), maximum curvature, mean curvature and ratio between maximum and mean curvatures. Mean values and coefficients of variation (given in parentheses) are indicated in the upper row, and the minimum and maximum values are shown in the lower row. Different superscript letters indicate significant differences (*p* < 0.05). N indicates the number of tubercles analyzed.

	N	W	H	Slope	Max Curvature	Mean Curvature	Max C/Mean C
Rugose	268	83.3 ^a^ (24.8) 31.1/174.0	20.7 ^b^ (35.5) 6.8/42.0	49.8 ^b^ (25.8) 22.5/89.7	41.4 ^b^ (47.1) 14.5/150.4	26.9 ^b^ (41.3) 9.4/69.0	1.6 ^b^ (30.8) 1.0/3.9
Echinate	128	72.0 ^b^ (23.1) 31.0/127.0	43.6 ^a^ (49.7) 13.0/100.0	124.3 ^a^ (52.2) 40.0/394.3	118.7 ^a^ (63.8) 17.3/477.0	44.2 ^a^ (48.3) 6.7/126.7	3.1 ^a^ (77.0) 0.27/18.4

**Table 2 plants-12-03444-t002:** Results of the comparison (ANOVA) between 21 species with rugose seeds for tubercle width (W), height (H), slope (S), maximum curvature, mean curvature and ratio between maximum and mean curvatures. Mean values and coefficients of variation (given in parentheses) are indicated in the upper row, and the minimum and maximum values are shown in the lower row. Mean values marked with different superscript letters indicate significant differences (*p* < 0.05). N indicates the number of tubercles analyzed (the number of seeds is indicated between parenthesis). A total of 268 analyzed tubercles are presented in this section.

Species	N	W	H	Slope	Max Curvature	Mean Curvature	Max C/Mean Curv
*S. acutifolia* (L.) Link ex Rohrb.	22 (2)	92.3 ^c^ (17.4) 68.6/127.0	28.0 ^d^ (21.1) 18.9/41.8	61.5 ^c^ (20.1) 41.9/89.7	51.0 ^abcd^ (10.9) 43.7/64.8	33.2 ^cdef^ (15.5) 17.9/42.7	1.6 ^abc^ (23.8) 1.1/3.0
*S. bupleuroides* L.	18 (2)	88.6 ^c^ (16.8) 53.2/115.5	20.8 ^abcd^ (27.9) 10.9/34.9	46.6 ^abc^ (15.6) 34.5/60.4	30.7 ^a^ (29.2) 17.0/49.2	18.5 ^abcd^ (27.5) 9.8/28.5	1.7 ^abc^ (17.9) 1.3/2.5
*S. caryophylloides* (Poir.) Otth	20 (2)	91.8 ^c^ (21.8) 60.4/129.0	26.4 ^cd^ (24.3) 12.7/40.0	57.4 ^bc^ (12.3) 42.0/71.1	40.3 ^abc^ (16.3) 23.7/50.0	32.9 ^bcdef^ (18.2) 19.9/42.0	1.2 ^a^ (10.8) 1.1/1.5
*S. chlorantha* (Willd.) Ehrh.	20 (2)	76.1 ^abc^ (19.5) 45.0/95.8	23.2 ^bcd^ (30.2) 12.0/36.9	60.3 ^c^ (19.2) 37.9/77.8	31.4 ^a^ (18.4) 22.2/43.0	18.9 ^abcd^ (19.7) 12.6/24.6	1.7 ^abc^ (23.7) 1.2/2.8
*S. chlorifolia* Sm.	10 (1)	98.5 ^c^ (10.7) 79.0/117.3	26.4 ^cd^ (18.5) 14.5/31.7	53.7 ^bc^ (18.3) 32.5/67.6	29.6 ^a^ (36.7) 17.8/57.2	15.1 ^ab^ (14.0) 11.2/17.9	1.9 ^abc^ (29.6) 1.2/3.2
*S. dinarica* Spreng.	18 (3)	86.1 ^bc^ (13.7) 64.8/108.4	20.4 ^abcd^ (18.2) 13.9/26.2	47.5 ^abc^ (16.4) 35.0/66.1	42.5 ^abc^ (34.2) 28.0/73.0	32.8 ^bcdef^ (26.9) 23.1/56.0	1.3 ^a^ (11.0) 1.1/1.6
*S. foliosa* Maxim.	16 (2)	85.2 ^bc^ (18.0) 60.8/108.0	26.2 ^cd^ (34.9) 13.6/42.0	60.7 ^c^ (23.6) 33.3/86.6	32.2 ^a^ (27.7) 20.6/55.2	21.8 ^abcde^ (23.4) 11.4/34.1	1.5 ^abc^ (20.5) 1.2/2.3
*S. frivaldskiana* Hampe	18 (2)	71.8 ^abc^ (12.5) 55.4/87.7	12.2 ^ab^ (21.9) 9.3/17.4	34.3 ^ab^ (23.5) 24.9/50.7	30.4 ^a^ (30.4) 19.5/56.2	18.8 ^abcd^ (22.0) 9.6/29.8	1.6 ^abc^ (21.4) 1.2/2.7
*S. gigantea* (L.) L	10 (1)	66.0 ^abc^ (8.9) 61.0/81.0	16.6 ^abcd^ (22.8) 14.0/27.0	50.5 ^abc^ (23.0) 39.5/80.6	51.3 ^abcd^ (59.1) 18.0/114.0	28.6 ^abcde^ (29.3) 14.0/37.0	1.8 ^abc^ (50.3) 1.1/3.9
*S. inaperta* L.	19 (2)	45.0 ^a^ (16.8) 31.1/63.0	9.5 ^a^ (18.2) 6.8/12.9	42.5 ^abc^ (15.4) 32.6/53.9	54.1 ^abcd^ (37.1) 18.0/112.0	39.6 ^ef^ (28.2) 14.0/58.0	1.4 ^abc^ (31.6) 1.1/2.8
*S. integripetala* Bory & Chaub.	13 (1)	86.4 ^bc^ (12.8) 61.0/104.9	22.5 ^abcd^ (12.5) 16.0/27.7	52.4 ^bc^ (11.1) 43.9/66.3	23.7 ^a^ (26.3) 14.5/39.4	16.7 ^abc^ (30.6) 10.0/28.0	1.4 ^a^ (11.7) 1.1/1.7
*S. koreana* Kom.	12 (1)	73.1 ^abc^ (19.3) 50.0/94.0	17.8 ^abcd^ (41.2) 8.0/31.0	46.9 ^abc^ (26.4) 24.2/66.7	32.9 ^a^ (31.9) 17.0/53.9	22.7 ^abcde^ (32.4) 9.4/34.5	1.4 ^abc^ (46.6) 1.0/3.7
*S. linicola* C.C. Gmel.	20 (2)	95.4 ^c^ (11.9) 80.0/115.0	22.9 ^abcd^ (17.3) 16.0/29.0	48.2 ^abc^ (17.5) 33.7/62.7	40.0 ^ab^ (27.2) 21.9/58.3	25.5 ^abcde^ (26.2) 16.7/41.1	1.6 ^abc^ (23.0) 1.2/2.6
*S. marizii* Samp.	6 (1)	138.7 ^d^ (15.0) 118.0/174.0	19.5 ^abcd^ (21.0) 15.0/26.0	28.0 ^a^ (8.9) 24.2/30.0	33.3 ^a^ (18.9) 26.0/43.0	20.2 ^abcd^ (37.0) 14.3/33.6	1.7 ^abc^ (15.9) 1.3/2.0
*S. nocturna* L.	6 (1)	54.7 ^ab^ (14.5) 45.0/64.0	13.5 ^abc^ (16.7) 10.0/16.0	49.3 ^abc^ (5.7) 44.4/53.1	30.3 ^a^ (17.4) 24.9/39.1	12.7 ^a^ (9.9) 10.6/14.0	2.4 ^bc^ (17.7) 2.0/3.0
*S. otites* (L.) Wibel	6 (1)	93.2 ^c^ (10.7) 78.0/109.0	16.2 ^abcd^ (32.8) 11.0/24.0	34.2 ^ab^ (24.3) 24.4/44.0	75.6 ^bcd^ (30.4) 46.5/103.9	47.7 ^f^ (33.0) 24.8/62.7	1.7 ^abc^ (29.2) 1.3/2.6
*S. paradoxa* L.	10 (1)	88.7 ^c^ (9.6) 78.0/101.0	23.2 ^bcd^ (14.6) 20.0/29.0	52.3 ^abc^ (12.1) 45.5/63.0	54.7 ^abcd^ (25.2) 45.0/80.0	35.3 ^def^ (4.4) 33.1/36.8	1.5 ^abc^ (22.3) 1.3/2.2
*S. pygmaea* Adams.	6 (1)	97.3 ^c^ (10.3) 82.0/108.0	20.2 ^abcd^ (29.3) 11.0/26.0	41.2 ^abc^ (24.7) 22.5/48.5	43.4 ^abc^ (19.6) 34.8/59.0	29.6 ^abcde^ (12.5) 23.8/34.3	1.5 ^abc^ (19.1) 1.2/1.9
*S. regia* Sims.	6 (1)	91.7 ^c^ (4.7) 86.0/96.0	16.2 ^abcd^ (23.9) 12.0/21.0	35.1 ^ab^ (20.9) 25.5/43.8	46.5 ^abc^ (9.5) 40.2/53.9	36.3 ^def^ (8.4) 32.3/40.5	1.3 ^ab^ (10.2) 1.2/1.4
*S. spinescens* Sm.	6 (1)	75.2 ^abc^ (4.4) 71.0/80.0	23.7 ^bcd^ (19.8) 18.0/30.0	63.4 ^c^ (22.5) 46.2/80.0	85.2 ^d^ (43.5) 47.3/150.4	48.1 ^f^ (33.0) 25.3/69.0	1.8 ^abc^ (16.7) 1.4/2.2
*S. squamigera* Boiss. subsp. *vesiculifera* (J.Gay ex Boiss.) Coode & Cullen	6 (1)	88.0 ^c^ (5.0) 81.0/92.0	18.8 ^abcd^ (14.0) 15.0/23.0	42.9 ^abc^ (13.6) 32.6/50.6	77.4 ^cd^ (56.6) 40.4/147.0	32.0 ^bcdef^ (30.2) 13.3/40.3	2.5 ^c^ (45.7) 1.4/3.9

**Table 3 plants-12-03444-t003:** Values of width (W), height (H), slope (S), maximum curvature, mean curvature and ratio of maximum curvature/mean curvature of the six tubercles of *Silene bupleuroides* numbered 1 to 6 in Figure 1.

Tubercle Number	1	2	3	4	5	6
W	87	68	91	89	91	90
H	15	17	21	23	20	16
S	34.5	50.0	46.2	51.7	44.0	35.6
Max curvature	32.1	45.7	29.8	24.2	36.1	28.4
Mean curvature	14.9	28.5	20.7	15.7	19.0	18.8
Ratio Max/Mean	2.2	1.6	1.4	1.5	1.9	1.5

**Table 4 plants-12-03444-t004:** Values of width (W), height (H), slope (S), maximum curvature, mean curvature and ratio of maximum curvature/mean curvature of the six tubercles of *Silene chlorantha* numbered 1 to 6 in Figure 2.

Tubercle Number	1	2	3	4	5	6
W	64	54	45	57	76	73
H	18	20	12	12	19	18
S	56.3	74.1	53.3	42.1	50.0	49.3
Max curvature	35.2	29.9	43.6	26.1	27.4	39.9
Mean curvature	20.9	19.0	19.6	13.6	12.6	16.8
Ratio Max/Mean	1.7	1.6	1.6	1.8	1.7	2.4

**Table 5 plants-12-03444-t005:** Values of width (W), height (H), slope (S), maximum curvature, mean curvature and ratio of maximum curvature/mean curvature in six tubercles of *S. frivaldskiana* numbered 1 to 6 in Figure 3.

Tubercle Number	1	2	3	4	5	6
W	65	73	83	63	77	74
H	11	9	14	15	10	12
S	32.9	25.5	34.3	47.0	26.7	31.5
Max curvature	35.7	32.1	31.5	22.6	22.0	28.7
Mean curvature	22.0	20.2	21.0	19.4	18.1	22.7
Ratio Max/Mean	1.6	1.6	1.5	1.2	1.2	1.3

**Table 6 plants-12-03444-t006:** Values of width (W), height (H), slope (S), maximum curvature, mean curvature and ratio of maximum curvature/mean curvature in six tubercles of *S. integripetala* numbered 1 to 6 in Figure 4.

Tubercle Number	1	2	3	4	5	6
W	61	81	88	96	98	79
H	16	22	25	23	22	22
S	52.5	54.3	56.8	47.9	44.9	55.7
Max curvature	39.4	23.9	25.4	23.2	20.9	23.9
Mean curvature	28.0	20.7	19.9	12.9	10.9	18.6
Ratio Max/Mean	1.4	1.2	1.3	1.8	1.9	1.3

**Table 7 plants-12-03444-t007:** Values of width (W), height (H), slope (S), maximum curvature, mean curvature and ratio maximum curvature/mean curvature in six tubercles of *S. spinescens* numbered 1 to 6 in Figure 5.

Tubercle Number	1	2	3	4	5	6
W	78	71	75	73	74	80
H	18	27	30	26	21	20
S	46.2	76.1	80.0	71.2	56.8	50.0
Max curvature	65.4	150.4	100.9	85.7	61.7	47.3
Mean curvature	35.9	69.0	55.1	57.8	45.2	25.3
Ratio Max/Mean	1.8	2.2	1.8	1.5	1.4	1.9

**Table 8 plants-12-03444-t008:** Values of width (W), height (H), slope (S), maximum curvature, mean curvature and ratio maximum curvature/mean curvature in six tubercles of *S. squamigera* subsp. *vesiculifera* numbered 1 to 6 in Figure 6.

Tubercle Number	1	2	3	4	5	6
W	92	81	91	86	86	92
H	20	18	23	18	19	15
S	43.5	44.4	50.5	41.9	44.2	32.6
Max curvature	52.1	57.6	97.3	147.0	117.4	89.7
Mean curvature	33.7	43.3	52.1	46.2	33.6	35.4
Ratio Max/Mean	1.5	1.3	1.9	3.2	3.5	2.5

**Table 9 plants-12-03444-t009:** Values of width (W), height (H), slope (S), maximum curvature, mean curvature and ratio maximum curvature/mean curvature in six tubercles of *S. gigantea* numbered 1 to 6 in Figure 7.

Tubercle Number	1	2	3	4	5	6
W	65	61	62	67	81	67
H	14	17	15	14	16	27
S	43.1	55.7	48.4	41.8	39.5	80.6
Max curvature	41.0	51.5	41.6	27.2	90.2	114.3
Mean curvature	31.1	37.2	31.4	21.7	32.5	29.5
Ratio Max/Mean	1.3	1.4	1.3	1.3	2.8	3.9

**Table 10 plants-12-03444-t010:** Results of the comparison (ANOVA) between 11 species of echinate seeds for tubercle width (W), height (H) and slope (S), maximum curvature, mean curvature and ratio of maximum and mean curvatures. Different letters in the superindex indicate significant differences (*p* < 0.05). N indicates the number of tubercles analyzed (the number of seeds is indicated between parentheses). A total of 128 tubercles were analyzed in this section.

Species	N	W	H	S	Max Curvature	Mean Curvature	Max C/Mean Curv
*S. aprica* subsp. *oldhamiana* (Miq.) C.Y.Wu ex C.L.Tang	12 (2)	61.0 ^ab^ (20.5) 42.2/77.0	49.3 ^bcd^ (15.5) 40.8/64.0	164.4 ^c^ (12.3) 135.5/206.6	104.2 ^abc^ (25.1) 77.0/167.7	38.4 ^ab^ (22.1) 27.0/58.0	2.9 ^ab^ (39.0) 1.7/5.4
*S. behen* L.	14 (2)	84.5 ^bc^ (15.7) 61.5/101.7	77.3 ^f^ (23.4) 42.0/100.0	182.8 ^c^ (18.8) 133.3/229.9	199.9 ^cd^ (67.5) 86.1/447.0	49.1 ^bc^ (15.0) 39.0/66.0	4.4 ^ab^ (79.5) 1.7/10.9
*S. caroliniana* Walter	22 (2)	72.5 ^bc^ (22.9) 53.0/108.1	32.8 ^ab^ (31.9) 15.0/56.0	90.6 ^ab^ (20.6) 40.5/123.1	108.5 ^abcd^ (24.3) 62.5/172.0	20.5 ^a^ (42.4) 6.7/41.3	6.1 ^b^ (52.1) 3.1/18.4
*S. ciliata* Pourr.	20 (2)	64.9 ^ab^ (15.7) 48.0/83.9	32.8 ^ab^ (21.6) 20.3/45.6	101.1 ^ab^ (15.4) 62.3/128.1	78.5 ^a^ (28.9) 42.0/126.0	38.7 ^ab^ (19.2) 28.7/57.7	2.1 ^a^ (27.4) 1.2/3.3
*S. dichotoma* Ehrh.	13 (2)	81.2 ^bc^ (14.3) 60.9/94.3	35.1 ^abc^ (18.8) 23.2/46.9	86.8 ^ab^ (15.3) 64.9/108.2	62.6 ^a^ (18.1) 49.2/84.1	40.6 ^ab^ (17.5) 32.7/53.8	1.6 ^a^ (14.7) 1.2/2.0
*S. fabaria* (L.) Coyte	6 (2)	84.5 ^bc^ (10.8) 74.0/97.0	71.2 ^ef^ (13.2) 59.0/83.0	168.3 ^c^ (6.8) 156.5/185.2	161.2 ^abcd^ (21.4) 110.0/202.0	78.5 ^de^ (13.1) 68.0/92.0	2.1 ^a^ (20.4) 1.6/2.8
*S. longicilia* (Brot.) Otth	25 (1)	70.9 ^bc^ (15.7) 42.0/87.0	23.7 ^a^ (30.9) 13.0/36.0	67.3 ^a^ (29.4) 40.0/120.0	81.5 ^ab^ (50.7) 17.3/167.0	40.5 ^ab^ (29.0) 20.2/64.1	2.0 ^a^ (33.6) 0.3/3.3
*S. petersonii* Maguire	4 (2)	68.0 ^abc^ (10.5) 61.0/78.0	83.3 ^f^ (13.4) 71.0/98.0	247.8 ^d^ (20.5) 212.8/321.3	191.5 ^bcd^ (11.7) 178.0/225.0	79.6 ^de^ (17.2) 60.3/89.6	2.4 ^ab^ (16.1) 2.1/3.0
*S. samojedorum* (Sambuk) Oxelman	6 (2)	91.2 ^c^ (25.5) 72.0/127.0	65.2 ^def^ (21.2) 52.0/87.0	146.6 ^bc^ (19.2) 100.0/175.0	219.2 ^d^ (37.6) 143.0/337.0	91.5 ^e^ (28.0) 71.0/126.7	2.4 ^a^ (25.1) 1.8/3.4
*S. yunnanensis* Franch	6 (1)	42.2 ^a^ (33.8) 31.0/66.0	54.2 ^cde^ (30.8) 27.0/71.0	278.5 ^d^ (38.9) 101.9/394.3	214.3 ^cd^ (31.8) 149.8/341.2	68.0 ^cd^ (32.4) 42.2/107.2	3.2 ^ab^ (18.3) 2.8/4.2

**Table 11 plants-12-03444-t011:** Values of width (W), height (H), slope (S), maximum curvature, mean curvature and ratio of maximum curvature/mean curvature in six tubercles of *S. aprica* numbered 1 to 6 in Figure 8.

Tubercle Number	1	2	3	4	5	6
W	75	76	77	73	62	54
H	59	64	55	53	42	43
S	157.3	168.4	142.9	145.2	135.5	159.3
Max curvature	120.5	160.5	80.9	88.6	123.5	90.2
Mean curvature	37.1	30.8	33.3	37.4	40.2	39.9
Ratio Max/Mean	3.2	5.2	2.4	2.4	3.1	2.3

**Table 12 plants-12-03444-t012:** Values of width (W), height (H), slope (S), maximum curvature, mean curvature and ratio of maximum curvature/mean curvature in six tubercles of *S. caroliniana* numbered 1 to 6 in Figure 9.

Tubercle Number	1	2	3	4	5	6
W	79	74	59	64	69	54
H	32	15	26	35	29	29
S	81.0	40.5	88.1	109.4	84.1	107.4
Max curvature	78.3	44.2	72.6	172.3	110.5	146.7
Mean curvature	23.6	17.7	29.9	37.7	25.2	31.9
Ratio Max/Mean	3.3	2.5	2.4	4.6	4.4	4.6

**Table 13 plants-12-03444-t013:** Values of width (W), height (H), slope (S), maximum curvature, mean curvature and ratio of maximum curvature/mean curvature in six tubercles of *S. ciliata* numbered 1 to 6 in Figure 10.

Tubercle Number	1	2	3	4	5	6
W	69	49	48	61	73	73
H	34	28	23	30	33	42
S	99.9	116.3	95.8	98.0	91.5	115.5
Max curvature	74.4	67.7	42.9	67.2	41.8	83.7
Mean curvature	43	58	34	45	32	44
Ratio Max/Mean	1.7	1.8	1.3	1.5	1.3	1.9

**Table 14 plants-12-03444-t014:** Mean values of width (W), height (H), slope (S), maximum curvature, mean curvature and ratio of maximum curvature/mean curvature in six tubercles of *S. yunnanensis* numbered 1 to 6 in Figure 11.

Tubercle Number	1	2	3	4	5	6
W	53	66	35	31	31	37
H	27	71	69	58	43	57
S	101.9	215.2	394.3	374.2	277.4	308.1
Max curvature	202.5	149.8	178.6	179.6	341.2	234.2
Mean curvature	72.5	54.5	42.2	65.0	107.2	66.5
Ratio Max/Mean	2.8	2.7	4.2	2.8	3.2	3.5

**Table 15 plants-12-03444-t015:** Results of the comparison (ANOVA) between 12 species of *Silene* subg. *Behenantha* and 19 species of *S.* subg. *Silene* for tubercle width (W), height (H) and slope (S), maximum curvature, mean curvature, and ratio of maximum and mean curvature. Different letters in the superindex indicate significant differences (*p* < 0.05). N indicates the number of tubercles analyzed.

	N	W	H	Slope	Max Curvature	Mean Curvature	Max/Mean Curvature
*Behenantha*	130	82.2 ^a^ (27.4) 31.0/174.0	41.9 ^a^ (53.4) 12.0/100.0	112.4 ^a^ (64.2) 24.2/394.3	103.8 ^a^ (79.4) 14.5/447.0	39.7 ^a^ (57.6) 6.7/126.7	2.9 ^a^ (84.2) 1.1/18.4
*Silene*	263	78.4 ^a^ (23.9) 31.1/129.0	21.4 ^b^ (37.9) 6.8/45.6	55.1 ^b^ (35.4) 22.4/128.1	48.2 ^b^ (57.1) 17.0/167.0	29.0 ^b^ (42.0) 9.4/69.0	1.7 ^b^ (33.3) 0.3/3.9

## Data Availability

Data are contained within the article or Appendix A.

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
