# Peer review of "Quantitative Analysis of Seed Surface Tubercles in Silene Species"

_plants, 2023, doi:10.3390/plants12193444_

Round 1
Reviewer 1 Report
Authors in the manuscript entitled “Quantitative analysis of seed surface tubercles in Silene species” examined tubercle width, height and slope, maximum and mean curvature values and maximum to mean curvature ratio for representative seeds of a total of 31 species, 21 of rugose seeds and 10 echinate. However, the manuscript in this version just presents basic data and doesn't reveal any new information. So the manuscript cannot be accepted in this current version.
Minor editing of English language is required.
Author Response
Dear Reviewer 1,
Thank you very much for your comments to our article.
According to the indication that the manuscript in the previous version only presented basic data without any new information, we present a revised version with modifications in all sections and a new sub-section (2.4) in the results section entitled: “The Relationship between tubercle type and subgenera”.
A new Table (Table 11) and a new Figure (Figure 15) have been added in the new sub-section with information showing the comparison of the tubercle measurements and curvature values in Silene subg. Silene and S. subg. Behenantha with a dendrogram based on the given measurements that shows the relationship between species in the two subgenera based on tubercle geometry. High curvature values, height and slope are characteristics of echinate tubercle geometry, and also of the tubercles of species in Silene subg. Behenantha in contrast with lower values in subg. Silene.
The text has been corrected putting more emphasis on the interest of the work in the introduction, discussion, and conclusions. Following your comments, the text has also been modified to include new information on the interest of the protocols described for the first time in this article as new contributions to seed morphology and taxonomy.
In the introduction the following sentence has been added:
In the scientific literature the description of tubercles is based on adjectives describing their main characteristics, but an analytical description based on measurements is lacking.
The discussion has been divided in two sections, dedicated respectively to the application of the protocols described in this article for the analysis of micromorphology in Silene seeds and in other Caryophyllaceae, and commenting on the differences described in the results between subgenera.
With the modifications done, we hope you may find the new protocols described in this article as a valuable new tool for the study of seed micromorphology and their application in taxonomy.
Looking forward to your answer, on behalf of the authors,
Emilio Cervantes
Corresponding author
Reviewer 2 Report
The article is scientifically relevant, well structured, and congruent among its different sections; however, it is not clear about the number of seeds used for analyzing the tubercles. It is quite clear the number of tubercles analyzed per species but, not at all, the number of seeds used for that. Hence, there is a doubt concerning the number of replicates used, or instead of replicates, there were subreplicates. Important to clarify this point.
There are few corrections to be done through the manuscript.
Please, see file attached.

Author Response
Dear Reviewer 2,
Thank you very much for your comments which have contributed significantly to improving the quality of our article.
The manuscript has been revised taking into account your comments. Statistical aspects have now been explained in more detail. The number of seeds analysed for each species is given in the materials and methods section, as well as in Tables 2 and 10 and in the legend to Figure A1 in Appendix A.
The following statements have been added:
In the Results section (2.1):
For each species, the number of seeds was comprised between 1 and 3, and a total of 49 seeds were analyzed (see Figure A1 in Appendix A).
The number of seeds has been now indicated between parenthesis in the legends to Tables 2 and 10, as well as in a new column in these two tables.
A new sub-section has been added to the Materials and methods section:
4.3. Measurements made on the tubercle images
Measurements of width, height, slope and curvature (maximum and mean curvature values) were made for a total of 396 tubercles (268 corresponding to rugose seeds and 128 to echinate seeds). The number of seeds ranged from 1 to 3 for each species as indicated in Tables 2 and 10 and Figure A1 in Appendix A.
In addition, a new sub-section (2.4) has been added to the results entitled: “Relationship between tubercle type and subgenera”. This sub-section contains a new Table (Table 11) and Figure (Figure 15) with information showing the comparison of the tubercle measurements and curvature values between species of subg. Silene and Behenantha and with a dendrogram based on the given measurements that shows the relationship between tubercle geometry and taxonomic groups.
All your commentaries made as additions to the PDF have been taken into consideration and the corresponding corrections made.
Looking forward to your answer, on behalf of the authors,
Emilio Cervantes
Corresponding author
Reviewer 3 Report
Dear authors,
Please see my comments in the file attached. The major comments are below:
(1) The references are not fully cited. There are some papers, e.g. Kravtsova (several articles) about the Silene seeds. The references are cited in Sukhorukov & al. (2018, Bot. J. Linn. Soc.) and id. (2023, Frontiers in Pl. Sci.).
(2) I think it is better to write an additional subheading in the Discussion section about the comparison of Silene seeds with other Caryophyllaceae and similarly looking seeds of Portulaca species. Are there significant differences between Silene and Portulaca (see different papers of G. Ocampo).
Best wishes,

Author Response
Dear Reviewer 3,
Thank you very much for your comments which have contributed significantly to improving the quality of our article.
Following your recommendation, references have been added to the discussion and this section has been divided in two sub-sections, entitled:
3.1. Seed surface morphology in Silene species
And:
3.2. Seed micromorphology in the Caryophyllaceae
The corrections indicated have been made and a new sub-section with a new table and figure has been added to the results section, entitled: 2.4. Relationship between tubercle type and subgenera.
In the new subsection, a new table (Table 11) and a figure (Figure 15) have been added with information showing the comparison of tubercle measurements and curvature values in subgenus Silene and subgenus Behenantha and with a dendrogram based on the given measurements that shows the relationship between tubercle geometry and taxonomic groups. High curvature values, height and slope are characteristics of echinate tubercle geometry, and of the tubercles of species in Silene subg. Behenantha in contrast with lower values in subg. Silene.
All your comments made as additions to the PDF have been taken into consideration and corrections made accordingly.
Awaiting your reply, on behalf of the authors
Emilio Cervantes
Corresponding author
Round 2
Reviewer 2 Report
Authors have taken into account all the comments, suggestions and questions done in the first version of the manuscript.
The new subsection and the dendrogram have really enriched the article.
Indeed, the article has improved.